# Byzantine Resilient Distributed Multi-Task Learning

**Jiani Li, Waseem Abbas, and Xenofon Koutsoukos**
Department of Electrical Engineering and Computer Science
Vanderbilt University, Nashville, TN, USA
{jiani.li, waseem.abbas, xenofon.koutsoukos}@vanderbilt.edu

## Abstract

Distributed multi-task learning provides significant advantages in multi-agent networks with heterogeneous data sources where agents aim to learn distinct but correlated models simultaneously. However, distributed algorithms for learning relatedness among tasks are not resilient in the presence of Byzantine agents. In this paper, we present an approach for Byzantine resilient distributed multi-task learning. We propose an efficient online weight assignment rule by measuring the accumulated loss using an agent's data and its neighbors' models. A small accumulated loss indicates a large similarity between the two tasks. In order to ensure the Byzantine resilience of the aggregation at a normal agent, we introduce a step for filtering out larger losses. We analyze the approach for convex models and show that normal agents converge resiliently towards the global minimum. Further, aggregation with the proposed weight assignment rule always results in an improved expected regret than the non-cooperative case. Finally, we demonstrate the approach using three case studies, including regression and classification problems, and show that our method exhibits good empirical performance for non-convex models, such as convolutional neural networks.

## 1 Introduction

Distributed machine learning models are gaining much attention recently as they improve the learning capabilities of agents distributed within a network with no central entity. In a distributed multi-agent system, agents interact with each other to improve their learning capabilities by leveraging the shared information via exchanging either data or models. In particular, agents that do not have enough data to build refined models or agents that have limited computational capabilities, benefit most from such cooperation. Distributed learning also addresses the single point of failure problem as well as scalability issues and is naturally suited to mobile phones, autonomous vehicles, drones, healthcare, smart cities, and many other applications [1, 2, 3, 4].

In networks with heterogeneous data sources, it is natural to consider the multi-task learning (MTL) framework, where agents aim to learn distinct but correlated models simultaneously [5]. Typically, prior knowledge of the relationships among models is assumed in MTL. The relationships among agents can be promoted via several methods, such as mean regularization, clustered regularization, low-rank and sparse structures regularization [6, 7, 8]. However, in real-world applications, such relationships are unknown beforehand and need to be estimated online from data. Learning similarities among tasks to promote effective cooperation is a primary consideration in MTL. There has been extensive work for learning the relationship matrix *centrally* by optimizing a global convex regularized function [9, 10, 11]. In contrast, this paper focuses on computationally efficient *distributed* learning of the relationship among agents that does not require optimizing a relationship matrix centrally [12, 13, 14, 15].

Although the distributed approach to learning and promoting similarities among neighbors from online data has many advantages, it is not resilient to Byzantine agents. Fault-tolerance for MTL

is discussed in [5], focusing on dropped nodes that occasionally stop sending information to their neighbors. In [16], the relationship promoted by measuring the quadratic distance between two model parameters for distributed MTL is shown to be vulnerable to gradient-based attacks, and a Byzantine resilient distributed MTL algorithm is proposed for regression problems to cope with such attacks. The proposed algorithm relies on a user-defined parameter $F$ to filter out information from $F$ neighbors in the aggregation step and is resilient to $F$ Byzantine neighbors, but requires exponential time with respect to the number of neighbors.

In this paper, we propose an *online weight adjustment rule* for MTL that is guaranteed to achieve resilient convergence for every normal agent using the rule. Compared to [16], the proposed method is suited for both regression and classification problems, is resilient to an arbitrary number of Byzantine neighbors (without the need to select a pre-defined parameter $F$ bounding the number of Byzantine neighbors), and has linear time complexity. To the best of our knowledge, this is the first solution that aims to address the Byzantine resilient cooperation in distributed MTL networks via a resilient similarity promoting method. We note that the proposed rule is not limited to the multi-task setting but can also be used for general distributed machine learning and federated learning systems to achieve resilient consensus. We list our contributions below.

- We propose an efficient Byzantine resilient online weight adjustment rule for distributed MTL. We measure similarities among agents based on the accumulated loss of an agent's data and the models of its neighbors. In each iteration, a normal agent computes the weights assigned to its neighbors in time that is linear in the size of its neighborhood and the dimension of the data.

- We show that aggregation with the proposed weight assignment rule always results in an improved expected regret than the non-cooperative case, and normal agents converge resiliently towards the global minimum. Even when all the neighbors are Byzantine, a normal agent can still resiliently converge to the global minimum bounded by the same expected regret as without any cooperation with other agents, achieving resilience to an arbitrary number of Byzantine agents.

- We conduct three experiments for both regression and classification problems and demonstrate that our approach yields good empirical performance for non-convex models, such as convolutional neural networks.

## 2 Related Work

**Multi-Task Learning.** MTL deals with the problem of learning multiple related tasks simultaneously to improve the generalization performance of the models learned by each task with the help of the other auxiliary tasks [17, 18]. The extensive literature in MTL can be broadly categorized into two categories based on how the data is collected. The *centralized* approach assumes the data is collected beforehand at a centralized entity. Many successful MTL applications with deep networks, such as in natural language processing and computer vision, fall into this category [19, 20, 21, 22]. This approach usually learns multiple objectives from a shared representation by sharing layers and splitting architecture in the deep networks. On the other hand, the *distributed* approach assumes data is collected separately by each task in a distributed manner. This approach is naturally suited to model distributed learning in multi-agent systems such as mobile phones, autonomous vehicles, and smart cities [2, 3, 4]. We focus on distributed MTL in this paper.

**Relationship Learning in MTL.** Although it is often assumed that a clustered, sparse, or low-rank structure among tasks is known *a priori* [6, 7, 8], such information may not be available in many real-world applications. Learning the relatedness among tasks online from data to promote effective cooperation is a principle approach in MTL when the relationships among tasks are not known *a priori*. There has been extensive work in online relationship learning that can be broadly categorized into centralized and distributed methods. The first group assumes that a centralized server collects the task models and utilizes a convex formulation of the regularized MTL optimization problem over the relationship matrix, which is learned by solving the convex optimization problem [9, 10, 11]. The second group relies on a distributed architecture in which agents learn relationships with their neighbors based on the similarities of their models and accordingly adjust weights assigned to neighbors [12, 13, 14, 15]. Typical similarity metrics, such as $\mathcal{H}$ divergence [23, 24, 25] and Wasserstein distance [25, 26], can be used in MTL in the same way they are used in domain adaptation, transfer learning, and adversarial learning. However, such metrics are mainly designed for measuring

the divergence in data distributions and are not suitable for online relationship learning due to efficiency and privacy concerns in data sharing.

**Resilient Aggregation in Distributed ML.** Inspired by the resilient consensus algorithms in multi-agent networks [27, 28], various resilient aggregation rules have been adapted in distributed ML, including the coordinate-wise trimmed mean [29], the coordinate-wise median [29, 30, 31], the geometric median [32, 33], and the Krum algorithm [34]. However, studies have shown that these rules are not resilient against certain attacks [35, 36, 37]. The centerpoint based aggregation rule [38] has been proposed recently that guarantees resilient distributed learning to Byzantine attacks. However, since each agent fits a distinct model in MTL, consensus-based resilient aggregation rules are not directly applicable to MTL.

## 3 Distributed Multi-Task Learning

**Notation.** In this paper, $|A|$ denotes the cardinality of a set $A$, $\|\cdot\|$ denotes the $\ell_2$ norm, $\text{Tr}(\cdot)$ denotes the trace of a matrix, and $\mathbb{E}_\xi[\cdot]$ denotes the expected value of a random variable $\xi$. If the context is clear, $\mathbb{E}[\cdot]$ is used.

**Background.** Consider a network of $n$ agents[1] modeled by an *undirected graph* $\mathcal{G} = (\mathcal{V}, \mathcal{E})$, where $\mathcal{V}$ represents agents and $\mathcal{E}$ represents interactions between agents. A bi-directional edge $(l, k) \in \mathcal{E}$ means that agents $k$ and $l$ can exchange information with each other. Since each agent also has its own information, we have $(k, k) \in \mathcal{E}, \forall k \in \mathcal{V}$. The *neighborhood* of $k$ is the set $\mathcal{N}_k = \{l \in \mathcal{V} | (l, k) \in \mathcal{E}\}$. Each agent $k$ has data $\{(x_k^i, y_k^i)\}$ sampled randomly from the distribution generated by the random variable $\xi_k$, where $x_k^i \in \mathbb{R}^{d_x}$, $y_k^i \in \mathbb{R}^{d_y}$. We use $\ell(\theta_k; \xi_k)$ to denote a convex *loss function* associated with the prediction function parameterized by $\theta_k$ for agent $k$. MTL is concerned with fitting separate models $\theta_k$ to the data for agent $k$ via the *expected risk function* $r_k(\theta_k) = \mathbb{E}[\ell(\theta_k; \xi_k)]$. We use $\theta_k^*$ to denote the global minimum of the convex function $r_k(\theta_k)$. The model parameters can be optimized via the following objective function:

$$\min_\Theta \left\{ \sum_{k=1}^n r_k(\theta_k) + \eta \mathcal{R}(\Theta, \Omega) \right\}, \tag{1}$$

where $\Theta = [\theta_1, \ldots, \theta_n] \in \mathbb{R}^{d_x \times n}$, $\mathcal{R}(\cdot)$ is a convex regularization function promoting the relationships among the agents, and $\Omega \in \mathbb{R}^{n \times n}$ models the relationships among the agents that can be assigned *a priori* or can be estimated from data. An example of the regularizer takes the form of $\mathcal{R}(\Theta, \Omega) = \lambda_1 \text{Tr}(\Theta \Omega \Theta^\top) + \lambda_2 \text{Tr}(\Theta \Theta^\top)$, where $\lambda_1, \lambda_2$ are non-negative parameters. In a *centralized* setting, where a centralized server optimizes the relationship matrix by collecting the models of agents, an optimal solution $\Omega = \frac{(\Theta^\top \Theta)^{\frac{1}{2}}}{\text{Tr}((\Theta^\top \Theta))^{\frac{1}{2}}}$ is proposed in [10] for learning the structure of clustered MTL using the above regularizer. In the *distributed* case, the task relationships $\Omega$ are not learned centrally and we can use the *adapt-then-combine (ATC) diffusion* algorithm [39] as a projection-based distributed solution of (1):

$$\hat{\theta}_{k,i} = \theta_{k,i-1} - \mu_k \nabla \ell(\theta_{k,i-1}; \xi_k^{i-1}), \qquad \text{(adaptation)} \tag{2}$$

$$\theta_{k,i} = \sum_{l \in \mathcal{N}_k} a_{lk} \hat{\theta}_{l,i}, \text{subject to} \sum_{l \in \mathcal{N}_k} a_{lk} = 1, a_{lk} \geq 0, a_{lk} = 0 \text{ if } l \notin \mathcal{N}_k, \quad \text{(combination)} \tag{3}$$

where $\mathcal{N}_k$ is the neighborhood of agent $k$, $\mu_k$ is the step size, and $a_{lk}$ denotes the weight assigned by agent $k$ to $l$, which should accurately reflect the similarity relationships among agents[2]. $\nabla \ell(\theta_{k,i-1}; \xi_k^{i-1})$ is the gradient using the instantaneous realization $\xi_k^{i-1}$ of the random variable $\xi_k$. At each iteration $i$, agent $k$ minimizes the individual risk using stochastic gradient descent (SGD) given local data followed by a combination step that aggregates neighboring models according to the weights assigned to them. The weights $\{a_{lk}\}$ are free parameters selected by the designer and they serve the same purpose as $\Omega$ in a centralized formulation. Thus, there is no need to design $\Omega$ in the case of distributed MTL that utilizes ATC diffusion algorithm for aggregation [40].

**Online Weight Adjustment Rules.** Without knowing the relationships *a priori*, one can assume the existence of similarities among agents and can learn these similarities online from data. The approach is based on the distance between the model parameters of agents, where a small distance indicates a large similarity [12, 13, 41, 42]. A common approach to learning similarities between two agents online is given by

$$a_{lk}(i) = \frac{\|\tilde{\theta}_k^* - \hat{\theta}_{l,i}\|^{-2}}{\sum_{p \in \mathcal{N}_k} \|\tilde{\theta}_k^* - \hat{\theta}_{p,i}\|^{-2}}, \tag{4}$$

where $\tilde{\theta}_k^*$ is an approximation of $\theta_k^*$ since $\theta_k^*$ is unknown. Examples include using the current model $\tilde{\theta}_k^* = \theta_{k,i-1}$, and one-step ahead approximation $\tilde{\theta}_k^* = \hat{\theta}_{k,i} + \mu_k \nabla \ell(\hat{\theta}_{k,i}; \xi_k^{i-1})$. Although the $\ell_2$ norm is widely used, this formulation of weights can be generalized to $\ell_p$ norm as well.

# 4 Problem Formulation

Byzantine agents can send arbitrary different information to different neighbors usually with a malicious goal of disrupting the network's convergence. It has been shown in [16] that normal agents assigning weights according to (4) are vulnerable to Byzantine agents. Particularly, by sending $\|\hat{\theta}_{b,i} - \tilde{\theta}_k^*\| \ll \|\hat{\theta}_{k,i} - \tilde{\theta}_k^*\|$, a Byzantine agent $b$ can gain a large weight from $k$ and continuously drive its normal neighbor $k$ towards a desired malicious point.

To address the vulnerabilities of the online weight adjustment rules derived from (4), this paper aims to design an efficient resilient online weight assignment rule in the presence of Byzantine agents for MTL. Let the *expected regret* $\mathbb{E}[r_k(\theta_{k,i}) - r_k(\theta_k^*)]$ be the value of the expected difference between the risk of $\theta_{k,i}$ and the optimal decision $\theta_k^*$. We aim to design weights $A_k = [a_{1k}, \dots, a_{nk}] \in \mathbb{R}^{1 \times n}$ for a normal agent $k$ that satisfy the following conditions:

**Resilient Convergence.** It must be guaranteed that using the computed weights $A_k$, every normal agent $k$ resiliently converges to $\theta_k^*$, even in the presence of Byzantine neighbors.

**Improved Learning Performance.** Cooperation among agents is meaningful only when it improves the learning performance. Hence, it is important to guarantee that for every normal agent, the combination step using the computed weights $A_k$ always results in an improved expected regret, even in the presence of Byzantine agents, i.e.,

$$\mathbb{E}[r_k(\theta_{k,i}) - r_k(\theta_k^*)] \leq \mathbb{E}[r_k(\hat{\theta}_{k,i}) - r_k(\theta_k^*)], \forall k \in \mathcal{N}^+, i \in \mathbb{N} \tag{5}$$

**Computational Efficiency.** At each iteration, a normal agent $k$ needs to compute the weights $A_k$ in time that is linear in the size of the neighborhood of $k$ and the dimension of the data, i.e., in $O(|\mathcal{N}_k|(d_x + d_y))$ time.

# 5 Loss-based Online Weight Adjustment

## 5.1 Weight Optimization

We follow a typical approach of learning the optimal weight adjustment rule [12, 13, 41, 42] in which the goal is to minimize the quadratic distance between the aggregated model $\theta_{k,i}$ and the true model $\theta_k^*$ over the weights, i.e., $\min_{A_k} \|\theta_{k,i} - \theta_k^*\|^2$. Using (3), we get an equivalent problem:

$$\min_{A_k} \left\| \sum_{l \in \mathcal{N}_k} a_{lk} \hat{\theta}_{l,i} - \theta_k^* \right\|^2, \text{ subject to } \sum_{l \in \mathcal{N}_k} a_{lk} = 1, a_{lk} \geq 0, a_{lk} = 0 \text{ if } l \notin \mathcal{N}_k,$$

where $\left\| \sum_{l \in \mathcal{N}_k} a_{lk} \hat{\theta}_{l,i} - \theta_k^* \right\|^2 = \sum_{l \in \mathcal{N}_k} \sum_{p \in \mathcal{N}_k} a_{lk} a_{pk} (\hat{\theta}_{l,i} - \theta_k^*)^\top (\hat{\theta}_{p,i} - \theta_k^*)$. As in a typical approximation approach, we consider

$$\left\| \sum_{l \in \mathcal{N}_k} a_{lk} \hat{\theta}_{l,i} - \theta_k^* \right\|^2 \approx \sum_{l \in \mathcal{N}_k} a_{lk}^2 \left\| \hat{\theta}_{l,i} - \theta_k^* \right\|^2. \tag{6}$$

The weight assignment rule (4) is an optimal solution of (6) using the approximation of $\theta_k^*$, which as we discuss above, can be easily attacked. To avoid the use of the distance between model parameters as a similarity measure, we introduce a *resilient* counterpart, which is the *accumulated loss* (or risk). Assume risk functions $r_k$ to be $m$-strongly convex[3], then it holds that

$$r_k(\hat{\theta}_{l,i}) - r_k(\theta_k^*) \geq \langle \nabla r_k(\theta_k^*), \hat{\theta}_{l,i} - \theta_k^* \rangle + \frac{m}{2} \|\hat{\theta}_{l,i} - \theta_k^*\|^2,$$

where $r_k(\hat{\theta}_{l,i}) = \mathbb{E}\left[\ell(\hat{\theta}_{l,i}; \xi_k)\right]$. Since $\nabla r_k(\theta_k^*) = 0$, we obtain

$$\|\hat{\theta}_{l,i} - \theta_k^*\|^2 \leq \frac{2}{m}\left(r_k(\hat{\theta}_{l,i}) - r_k(\theta_k^*)\right). \tag{7}$$

Instead of directly minimizing the right side of (6), we consider minimizing its upper bound given in (7). Later in Section 6, we show that this alternate approach facilitates the resilient distributed MTL, which cannot be achieved by minimizing the distance between models directly. Hence, by combining (6) and (7), we consider the following minimization problem:

$$\min_{A_k} \sum_{l \in \mathcal{N}_k} a_{lk}^2 \left(r_k(\hat{\theta}_{l,i}) - r_k(\theta_k^*)\right) \text{ subject to } \sum_{l \in \mathcal{N}_k} a_{lk} = 1, a_{lk} \geq 0, a_{lk} = 0 \text{ if } l \notin \mathcal{N}_k.$$

This optimization problem indicates that if a neighbor $l$'s model has a small regret on agent $k$'s data distribution, then it should be assigned a large weight. Since $\theta_k^*$ is unknown, one can use $r_k(\theta_{k,i})$ to approximate $r_k(\theta_k^*)$. Alternatively, since $r_k(\theta_k^*)$ is small compared to $r_k(\hat{\theta}_{l,i})$, we could simply assume $r_k(\theta_k^*) = 0$ and consider the following minimization problem:

$$\min_{A_k} \sum_{l \in \mathcal{N}_k} a_{lk}^2 r_k(\hat{\theta}_{l,i}) \text{ subject to } \sum_{l \in \mathcal{N}_k} a_{lk} = 1, a_{lk} \geq 0, a_{lk} = 0 \text{ if } l \notin \mathcal{N}_k. \tag{8}$$

Using the Lagrangian relaxation, we obtain the optimal solution[4] of (8) as

$$a_{lk}(i) = \frac{r_k(\hat{\theta}_{l,i})^{-1}}{\sum_{p \in \mathcal{N}_k} r_k(\hat{\theta}_{p,i})^{-1}}. \tag{9}$$

We can approximate $r_k(\hat{\theta}_{l,i})$ using the exponential moving average $\varphi_{lk}^i = (1 - \nu_k)\varphi_{lk}^{i-1} + \nu_k \ell(\hat{\theta}_{l,i}; \xi_k)$, where $\nu_k$ is the forgetting factor. Given $\mathbb{E}[\varphi_{lk}^i] = (1 - \nu_k)\mathbb{E}[\varphi_{lk}^{i-1}] + \nu_k \mathbb{E}[\ell(\hat{\theta}_{l,i}; \xi_k)]$, we obtain $\lim_{i \to \infty} \mathbb{E}[\varphi_{lk}^i] = \lim_{i \to \infty} \mathbb{E}[\ell(\hat{\theta}_{l,i}; \xi_k)] = \lim_{i \to \infty} r_k(\hat{\theta}_{l,i})$, which means $\varphi_{lk}^i$ converges (in expectation) to $\lim_{i \to \infty} r_k(\hat{\theta}_{l,i})$. Hence, we can use $\varphi_{lk}^i$ to approximate $r_k(\hat{\theta}_{l,i})$. Note that in addition to the smoothing methods, one can use the average batch loss to approximate $r_k(\hat{\theta}_{l,i})$ when using the (mini-) batch gradient descent in the place of SGD for adaptation.

## 5.2 Filtering for Resilience

Let $\mathcal{N}_k^+$ denote the set of $k$'s normal neighbors with $|\mathcal{N}_k^+| \geq 1$. We assume there are $q$ Byzantine neighbors in the set $\mathcal{B} = \mathcal{N}_k \backslash \mathcal{N}_k^+$. In the following, we examine the resilience of the cooperation using (9) in the presence of Byzantine agents.

**Lemma 1.** [5] *The following condition holds for the combination step* (3) *using weights* (9)*:*

$$\mathbb{E}\left[r_k(\theta_{k,i}) - r_k(\theta_k^*)\right] \leq \frac{1}{|\mathcal{N}_k|} \sum_{l \in \mathcal{N}_k} \mathbb{E}\left[r_k\left(\hat{\theta}_{l,i}\right) - r_k(\theta_k^*)\right].$$

Since $l$ can be a Byzantine agent, it is possible that $\mathbb{E}\left[r_k\left(\hat{\theta}_{l,i}\right) - r_k(\theta_k^*)\right]$ is a large value. Consequently, we cannot compute a useful upper bound on the value of $\mathbb{E}\left[r_k(\theta_{k,i}) - r_k(\theta_k^*)\right]$ given Lemma 1 and cannot provide further convergence guarantees. To facilitate the resilient cooperation, we consider a modification of (9) as follows.

$$a_{lk}(i) = \begin{cases} \frac{r_k(\hat{\theta}_{l,i})^{-1}}{\sum_{p \in \mathcal{N}_k^{\leq}} r_k(\hat{\theta}_{p,i})^{-1}}, & \text{if } r_k(\hat{\theta}_{l,i}) \leq r_k(\hat{\theta}_{k,i}), \\ 0, & \text{otherwise,} \end{cases} \tag{10}$$

where $\mathcal{N}_k^{\leq}$ denotes the set of neighbors with $r_k(\hat{\theta}_{l,i}) \leq r_k(\hat{\theta}_{k,i})$. This implies that the cooperation filters out the information coming from the neighbors incurring a larger risk and cooperate only with the remaining neighbors. In the next section, we show how this modification benefits learning and guarantees the resilient convergence of MTL.

### 5.3 Computational Complexity

It takes $\mathcal{O}(d_x + d_y)$ time to compute $\ell(\hat{\theta}_{l,i}; \xi_k^i)$. Using the exponential moving average method for approximating $r_k(\hat{\theta}_{l,i})$, for a normal agent $k$, at each iteration $i$, the total time for computing $A_k(i)$ with the proposed rule (10) is $\mathcal{O}(|\mathcal{N}_k|(d_x + d_y))$.

## 6 Byzantine Resilient Convergence Analysis

We make the following general assumptions for the convergence of SGD [43] to derive our results.

**Assumption 1.** *For every normal agent $k$, the risk function $r_k(\cdot)$ is $m$-strongly convex and has $L$-Lipschitz continuous gradient.*[6]

**Assumption 2.** *For every normal agent $k$, the stochastic gradient $\nabla \ell(\theta_{k,i}; \xi_k^i)$ is an unbiased estimate of $\nabla r_k(\theta_{k,i})$, i.e., $\mathbb{E}[\nabla \ell(\theta_{k,i}; \xi_k^i)] = \nabla r_k(\theta_{k,i})$, for all $i \in \mathbb{N}$.*

**Assumption 3.** *For every normal agent $k$, there exists $c_k \geq 1$, such that for all $i \in \mathbb{N}$, $\mathbb{E}[\|\nabla \ell(\theta_{k,i}; \xi_k^i)\|_2^2] \leq \sigma_k^2 + c_k \|\nabla r_k(\theta_{k,i})\|_2^2$.*

Given these assumptions, the convergence of a normal agent running SGD is guaranteed with appropriate step size [43]. Using the proposed rule (10), under these assumptions, we further guarantee the convergence of the normal agents running the ATC diffusion algorithm in Theorem 1.

**Theorem 1.** *A normal agent $k$ which runs the ATC diffusion algorithm using the loss-based weights* (10) *converges towards $\theta_k^*$ with $\lim_{i \to \infty} \mathbb{E}\left[r_k(\theta_{k,i}) - r_k(\theta_k^*)\right] \leq \frac{\mu_k L \sigma_k^2}{2m}$, for fixed stepsize $\mu_k \in (0, \frac{1}{Lc_k}]$, in the presence of an arbitrary number of Byzantine neighbors. Further, it holds that $\mathbb{E}[r_k(\theta_{k,i}) - r_k(\theta_k^*)] \leq \mathbb{E}[r_k(\hat{\theta}_{k,i}) - r_k(\theta_k^*)], \forall k \in \mathcal{N}^+, i \in \mathbb{N}$.*

Theorem 1 indicates that cooperation using weights in (10) is always at least as good as the non-cooperative case, as measured by the expected regret, which satisfies the conditions lsited in Section 4. Note that even when all the neighbors of a normal agent are Byzantine, one can still guarantee that the agent's learning performance as a result of cooperation with neighbors using (10) will be same as the non-cooperative case.

**Discussion.** We assume convex models to carry out the analysis, which is typical in the literature. However, the intuition behind the approach is — *to measure the relatedness of a neighbor to itself, a normal agent evaluates the loss of the neighbor using the neighbor's model parameters and its own data, and cuts down the cooperation if this loss is larger than the agent's own loss* — and the same idea should also apply to non-convex models. In the next section, we also evaluate our methods on non-convex models, such as CNNs, which generates experimental results similar to those produced by convex models.

## 7 Evaluation

In this section, we evaluate the resilience of the proposed online weight adjustment rule (10) with the smoothing method discussed in Section 5, and compare it with the non-cooperative case, the average weights ($a_{lk} = \frac{1}{|\mathcal{N}_k|}$), and the quadratic distance-based weights (4) (with $\tilde{\theta}_k^* = \theta_{k,i-1}$ and use the same smoothing method $\phi_{lk}^i = (1 - \nu_k)\phi_{lk}^{i-1} + \nu_k \|\tilde{\theta}_k^* - \hat{\theta}_{l,i}\|^2$ in the place of $\|\tilde{\theta}_k^* - \hat{\theta}_{l,i}\|^2$, with the same forgetting factor $\nu_k$ used for (10)). We use three distributed MTL case studies, including the regression and classification problems, with and without the presence of Byzantine agents. Although the convergence analysis in Section 6 is based on convex models and SGD, we show empirically that the weight assignment rule (10) performs well for non-convex models, such as CNNs and mini-batch gradient descent. Our code is available at `https://github.com/JianiLi/resilientDistributedMTL`.

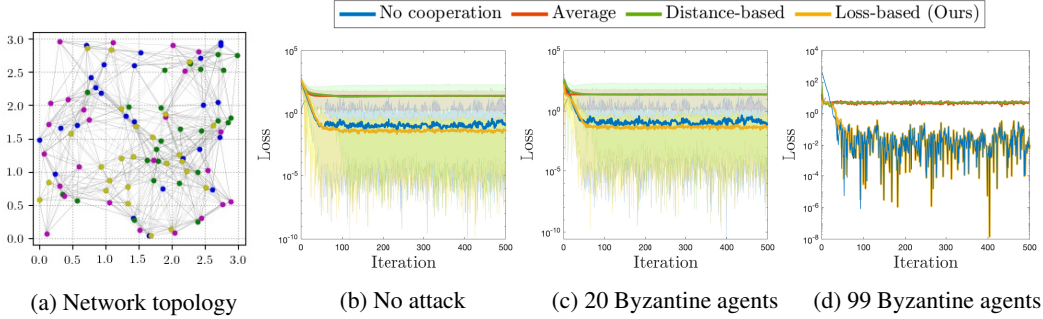

Figure 1: Target Localization: network topology and loss of streaming data for normal agents.

(a) Network topology   (b) No attack   (c) 20 Byzantine agents   (d) 99 Byzantine agents

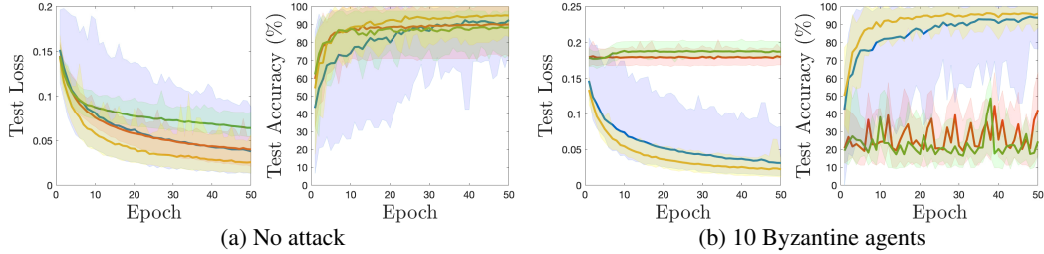

(a) No attack   (b) 10 Byzantine agents

Figure 2: Human Action Recognition: average testing loss and accuracy for normal agents.

## 7.1 Datasets and Simulation Setups

- **Target Localization**: Target localization is a widely-studied linear regression problem [44]. The task is to estimate the location of the target by minimizing the squared error loss of noisy streaming sensor data. We consider a network of 100 agents with four targets as shown in Figure 1a. Agents in the same color share the same target, however, they do not know this group information beforehand.

- **Human Activity Recognition**[7]: Mobile phone sensor data (accelerometer and gyroscope) is collected from 30 individuals performing one of six activities: {walking, walking-upstairs, walking-downstairs, sitting, standing, lying-down}. The goal is to predict the activities performed using 561-length feature vectors for each instance generated by the processed sensor signals [2]. We model each individual as a separate task and use a complete graph to model the network topology. We use linear model as the prediction function with cross-entropy-loss.

- **Digit Classification**: We consider a network of ten agents performing digit classification. Five of the ten agents have access to the MNIST dataset[8] [45] (group 1) and the other five have access to the synthetic dataset[9] (group 2) that is composed by generated images of digits embedded on random backgrounds [46]. All the images are preprocessed to be $28 \times 28$ grayscale images. We model each agent as a separate task and use a complete graph to model the network topology. An agent does not know which of its neighbors are performing the same task as the agent itself. We use a CNN model of the same architecture for each agent and cross-entropy-loss.

## 7.2 Results[10]

We plot the mean and range of the average loss of every normal agent for the target localization problem in Figure 1b–d. Similarly, we plot the mean and range of the average testing loss and classification accuracy of every normal agent for human action recognition in Figure 2, and for digit classification in Figure 3 (for group 1) and Figure 4 (for group 2). At each iteration, Byzantine agents

send random values (for each dimension) from the interval $[15, 16]$ for target localization, and $[0, 0.1]$ for the other two case studies.

In all of the examples, we find that the loss-based weight assignment rule (10) outperforms all the other rules and the non-cooperative case, with respect to the mean and range of the average loss and accuracy with and without the presence of Byzantine agents. Hence, our simulations validate the results indicated by (5) and imply that the loss-based weights (10) have accurately learned the relationship among agents. Moreover, normal agents having a large regret in their estimation benefit from cooperating with other agents having a small regret. We also consider the extreme case in which there is only one normal agent in the network, and all the other agents are Byzantine. In such a case, the loss-based weight assignment rule (10) has the same performance as the non-cooperative case, thus, showing that it is resilient to an arbitrary number of Byzantine agents.

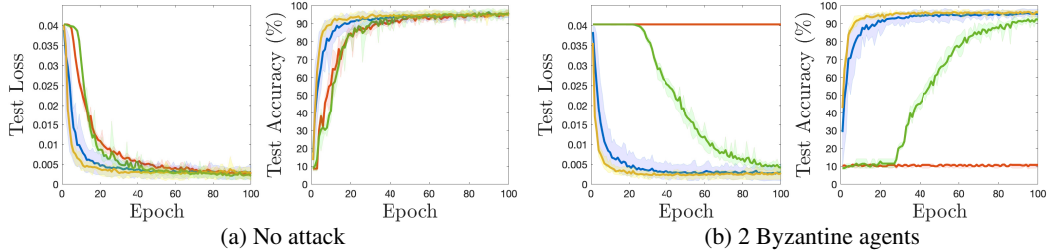

(a) No attack                                   (b) 2 Byzantine agents

Figure 3: Digit Classification: average testing loss and accuracy for normal agents in group 1.

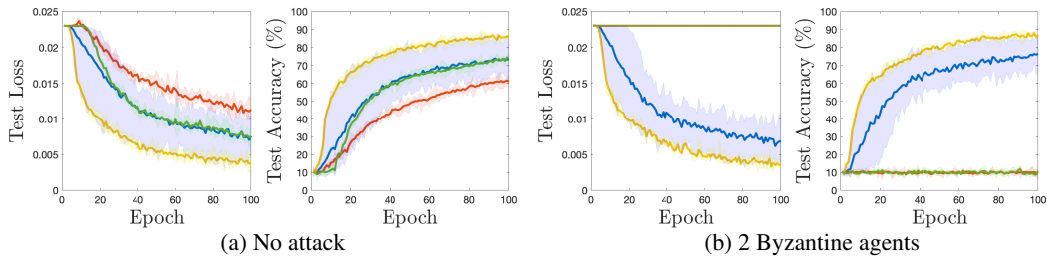

(a) No attack                                   (b) 2 Byzantine agents

Figure 4: Digit Classification: average testing loss and accuracy for normal agents in group 2.

## 8 Conclusion

In this paper, we propose an efficient online weight adjustment rule for learning the similarities among agents in distributed multi-task networks with an arbitrary number of Byzantine agents. We argue that a widely used approach of measuring the similarities based on the distance between two agents' model parameters is vulnerable to Byzantine attacks. To cope with such vulnerabilities, we propose to measure similarities based on the (accumulated) loss using an agent's data and its neighbors' models. A small loss indicates a large similarity between the agents. To eliminate the influence of Byzantine agents, a normal agent filters out the information from neighbors whose losses are larger than the agent's own loss. With filtering, aggregation using the loss-based weight adjustment rule results in an improved expected regret than the non-cooperative case and guarantees that each normal agent converges resiliently towards the global minimum. The experiment results validate the effectiveness of our approach.

## Broader Impact

The problem of Byzantine resilient aggregation of distributed machine learning models has been actively studied in recent years; however, the issue of Byzantine resilient distributed learning in multi-task networks has received much less attention. It is a general intuition that MTL is robust and resilient to cyber-attacks since it can identify attackers by measuring similarities between neighbors. In this paper, we have shown that some commonly used similarity measures are not resilient against certain attacks. With an increase in data heterogeneity, we hope this work could highlight the security and privacy concerns in designing distributed MTL frameworks.

## Acknowledgments and Disclosure of Funding

This work is supported in part by the NSA Lablet (H98230-18-D-0010). Any opinions, findings, and conclusions or recommendations expressed in this material are those of the author(s) and do not necessarily reflect the views of NSA.

## Footnotes

[1]Each agent is modeled as a separate task, thus, the terms *agent* and *task* are used interchangeably.

[2]$\mu_k$ and $a_{lk}$ can be time-dependent, but when context allows, we write $\mu_{k,i}$ as $\mu_k$ and $a_{lk}(i)$ as $a_{lk}$ for simplicity.

[3]Details of the assumptions are given in Appendix A.1.

[4]Detailed solution is given in Appendix A.2.

[5]All proofs are given in Appendix A; appendices can be found in the supplementary material.

[6]Details of the assumptions about the risk functions are given in Appendix A.1.

[7]https://archive.ics.uci.edu/ml/datasets/human+activity+recognition+using+smartphones

[8]http://yann.lecun.com/exdb/mnist

[9]https://www.kaggle.com/prasunroy/synthetic-digits

[10]Simulation details and supplementary results are given in Appendix B.

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
