[Supplementary Material]

# A Assumptions and Theoretical Results

## A.1 Assumptions of risk functions

**Definition 1.** *(L-Lipschitz continuous gradient). A differentiable convex function $f$ is said to have an L-Lipschitz continuous gradient, if there exists a constant $L > 0$, such that*

$$\|\nabla f(x) - \nabla f(y)\| \le L\|x - y\|, \forall x, y.$$

*If $f$ has an L-Lipschitz continuous gradient, then it holds that*

$$f(y) \le f(x) + \langle \nabla f(x), y - x \rangle + \frac{L}{2}\|y - x\|^2, \forall x, y.$$

**Definition 2.** *(m-strongly convex). A differentiable convex function $f$ is said to be m-strongly convex if there exists a constant $m > 0$, such that*

$$f(y) \ge f(x) + \langle \nabla f(x), y - x \rangle + \frac{m}{2}\|y - x\|^2, \forall x, y.$$

*If $f$ is m-strongly convex and has an L-Lipschitz continuous gradient, then it is obvious that $m \le L$.*

## A.2 Optimal solution of equation (8)

Let $\lambda$ be the Lagrange multiplier. We define the Lagrangian of (8) given the constraints on the weights as

$$\mathcal{L}(a_{lk}, \lambda) = \sum_{l \in \mathcal{N}_k} a_{lk}^2 r_k(\hat{\theta}_{l,i}) + \lambda(1 - \sum_{l \in \mathcal{N}_k} a_{lk}).$$

Set $\nabla_{a_{lk}, \lambda} \mathcal{L}(a_{lk}, \lambda) = \left( \frac{\partial \mathcal{L}}{\partial a_{lk}}, \frac{\partial \mathcal{L}}{\partial \lambda} \right) = 0$, i.e.,

$$\begin{cases} 2a_{lk} r_k(\hat{\theta}_{l,i}) - \lambda = 0, \forall l \in \mathcal{N}_k, \\ 1 - \sum_{l \in \mathcal{N}_k} a_{lk} = 0. \end{cases}$$

Thus, $a_{lk} = \frac{\lambda}{r_k(\hat{\theta}_{l,i})}, \forall l \in \mathcal{N}_k$ and $\sum_{l \in \mathcal{N}_k} a_{lk} = 1$. We have $\lambda \sum_{l \in \mathcal{N}_k} \frac{1}{r_k(\hat{\theta}_{l,i})} = 1$ and hence $\lambda = \frac{1}{\sum_{l \in \mathcal{N}_k} r_k(\hat{\theta}_{l,i})^{-1}}$, and $a_{lk} = \frac{r_k(\hat{\theta}_{l,i})^{-1}}{\sum_{p \in \mathcal{N}_k} r_k(\hat{\theta}_{p,i})^{-1}}$ is the optimal solution of (8).

## A.3 Proof of Lemma 1

*Proof.* Given (3), $r_k(\theta_{k,i}) = r_k \left( \sum_{l \in \mathcal{N}_k} a_{lk}(i)\hat{\theta}_{l,i} \right)$. Using Jensen's inequality, we have

$$r_k(\theta_{k,i}) \le \sum_{l \in \mathcal{N}_k} a_{lk}(i) r_k \left( \hat{\theta}_{l,i} \right). \tag{11}$$

Subtracting $r_k(\theta_k^*)$ from both sides of (11) and taking expectations over the joint distribution $\xi_k$, we obtain

$$\mathbb{E}\left[ r_k(\theta_{k,i}) - r_k(\theta_k^*) \right] \le \sum_{l \in \mathcal{N}_k} \mathbb{E}[a_{lk}(i)] \mathbb{E}\left[ r_k\left(\hat{\theta}_{l,i}\right) - r_k(\theta_k^*) \right]$$
$$\le \frac{\sum_{l \in \mathcal{N}_k} \mathbb{E}\left[ r_k(\hat{\theta}_{l,i}) \right]^{-1} \mathbb{E}\left[ r_k\left(\hat{\theta}_{l,i}\right) - r_k(\theta_k^*) \right]}{\sum_{p \in \mathcal{N}_k} \mathbb{E}\left[ r_k(\hat{\theta}_{p,i}) \right]^{-1}}. \tag{12}$$

We next prove the right-hand side of (12) is less than $\frac{1}{|\mathcal{N}_k|} \sum_{l \in \mathcal{N}_k} \mathbb{E}\left[ r_k(\hat{\theta}_{l,i}) - r_k(\theta_k^*) \right]$. For succinctness, we use $\chi_{l,i}$ to denote $\mathbb{E}\left[ r_k\left(\hat{\theta}_{l,i}\right) \right]^{-1}$, and $\Delta_{l,i}$ to denote $\mathbb{E}\left[ r_k\left(\hat{\theta}_{l,i}\right) - r_k(\theta_k^*) \right]$. And we aim to prove $\frac{\sum_{l \in \mathcal{N}_k} \chi_{l,i}\Delta_{l,i}}{\sum_{p \in \mathcal{N}_k} \chi_{p,i}} \le \frac{1}{|\mathcal{N}_k|} \sum_{l \in \mathcal{N}_k} \Delta_{l,i}$, or equivalently, $|\mathcal{N}_k| \sum_{l \in \mathcal{N}_k} \chi_{l,i}\Delta_{l,i} \le \sum_{p \in \mathcal{N}_k} \chi_{p,i} \sum_{l \in \mathcal{N}_k} \Delta_{l,i}$.

When $|\mathcal{N}_k| = 1$, one can easily validate that this condition holds. When $|\mathcal{N}_k| \ge 2$, let $l_1^i$ be the one with the smallest risk $r_k \left( \hat{\theta}_{l_1^i,i} \right) = \min_{l \in \mathcal{N}_k} r_k \left( \hat{\theta}_{l,i} \right)$ and $l_2^i$ be the one with the second smallest risk $r_k \left( \hat{\theta}_{l_2^i,i} \right) =$

$\min_{l \in \mathcal{N}_k \setminus l_1^i} r_k\left(\hat{\theta}_{l,i}\right)$. Hence, $\chi_{l_1^i,i} \geq \chi_{l_2^i,i} \geq \chi_{l,i}$, and $\Delta_{l_1^i,i} \leq \Delta_{l_2^i,i} \leq \Delta_{l,i}$ for $l \in \mathcal{N}_k \setminus \{l_1^i, l_2^i\}$. Thus,

$$|\mathcal{N}_k| \sum_{l \in \mathcal{N}_k} \chi_{l,i}\Delta_{l,i} - \sum_{p \in \mathcal{N}_k} \chi_{p,i} \sum_{l \in \mathcal{N}_k} \Delta_{l,i}$$

$$= \sum_{l \in \mathcal{N}_k} \chi_{l,i}\left(|\mathcal{N}_k|\Delta_{l,i} - \sum_{p \in \mathcal{N}_k} \Delta_{p,i}\right)$$

$$= \chi_{l_1^i,i}\left((|\mathcal{N}_k|-1)\Delta_{l_1^i,i} - \sum_{l \in \mathcal{N}_k \setminus l_1^i} \Delta_{l,i}\right) + \sum_{l \in \mathcal{N}_k \setminus l_1^i,i} \chi_{l,i}\left(|\mathcal{N}_k|\Delta_{l,i} - \sum_{p \in \mathcal{N}_k} \Delta_{p,i}\right)$$

$$\leq \chi_{l_1^i,i}\left((|\mathcal{N}_k|-1)\Delta_{l_1^i,i} - \sum_{l \in \mathcal{N}_k \setminus l_1^i} \Delta_{l,i}\right) + \chi_{l_2^i,i}\left(\sum_{l \in \mathcal{N}_k \setminus l_1^i} |\mathcal{N}_k|\Delta_{l,i} - (|\mathcal{N}_k|-1)\sum_{p \in \mathcal{N}_k} \Delta_{p,i}\right)$$

$$= \chi_{l_1^i,i}\left((|\mathcal{N}_k|-1)\Delta_{l_1^i,i} - \sum_{l \in \mathcal{N}_k \setminus l_1^i} \Delta_{l,i}\right) + \chi_{l_2^i,i}\left(\sum_{l \in \mathcal{N}_k \setminus l_1^i} \Delta_{l,i} - (|\mathcal{N}_k|-1)\Delta_{l_1^i,i}\right)$$

$$= \left(\chi_{l_1^i,i} - \chi_{l_2^i,i}\right)\left((|\mathcal{N}_k|-1)\Delta_{l_1^i,i} - \sum_{l \in \mathcal{N}_k \setminus l_1^i} \Delta_{l,i}\right)$$

$$= \left(\chi_{l_1^i,i} - \chi_{l_2^i,i}\right)\left(\sum_{l \in \mathcal{N}_k \setminus l_1^i} \left(\Delta_{l_1^i,i} - \Delta_{l,i}\right)\right) \leq 0.$$

Therefore, $\frac{\sum_{l \in \mathcal{N}_k} \chi_{l,i}\Delta_{l,i}}{\sum_{p \in \mathcal{N}_k} \chi_{p,i}} \leq \frac{1}{|\mathcal{N}_k|}\sum_{l \in \mathcal{N}_k}\Delta_{l,i}$. Put it back to (12), we obtain

$$\mathbb{E}\left[r_k(\theta_{k,i}) - r_k(\theta_k^*)\right] \leq \frac{1}{|\mathcal{N}_k|}\sum_{l \in \mathcal{N}_k} \mathbb{E}\left[r_k\left(\hat{\theta}_{l,i}\right) - r_k(\theta_k^*)\right],$$

which completes the proof. $\qquad\square$

### A.4 Proof of Theorem 1

*Proof.* Let $\mathbb{E}[\cdot]$ denote the expected value taken with respect to the joint distribution of all random variables $\xi_k$ and $\xi_l$ for $l \in \mathcal{N}_k^{\leq}$, i.e.
$$\mathbb{E}[\cdot] = \mathbb{E}_{\xi_k}\mathbb{E}_{\{\xi_l | l \in \mathcal{N}_k^{\leq}\}}[\cdot].$$

Similar to the proof for Lemma 1, using $\mathcal{N}_k^{\leq}$ in the place of $\mathcal{N}_k$, with rule (10), we obtain

$$\mathbb{E}\left[r_k(\theta_{k,i}) - r_k(\theta_k^*)\right] \leq \frac{1}{|\mathcal{N}_k^{\leq}|}\sum_{l \in \mathcal{N}_k} \mathbb{E}\left[r_k\left(\hat{\theta}_{l,i}\right) - r_k(\theta_k^*)\right]. \tag{13}$$

For every $l \in \mathcal{N}_k^{\leq}$, we have $r_k(\hat{\theta}_{l,i}) \leq r_k(\hat{\theta}_{k,i})$ and hence $\frac{1}{|\mathcal{N}_k^{\leq}|}\sum_{l \in \mathcal{N}_k} \mathbb{E}\left[\left(r_k(\hat{\theta}_{l,i}) - r_k(\theta_k^*)\right)\right] \leq \mathbb{E}\left[\left(r_k(\hat{\theta}_{k,i}) - r_k(\theta_k^*)\right)\right]$. Put it back to (13), we obtain

$$\mathbb{E}\left[r_k\left(\theta_{k,i}\right) - r_k(\theta_k^*)\right] \leq \frac{1}{|\mathcal{N}_k^{\leq}|}\sum_{l \in \mathcal{N}_k^{\leq}} \mathbb{E}\left[r_k\left(\hat{\theta}_{l,i}\right) - r_k(\theta_k^*)\right] \leq \mathbb{E}\left[r_k\left(\hat{\theta}_{k,i}\right) - r_k(\theta_k^*)\right], \forall k \in \mathcal{N}^+, i \in \mathbb{N},$$
$$\tag{14}$$

which yields (5).

We next prove the convergence of the algorithm with the proposed weight assignment rule. Given Assumptions 1-3, we obtain from [43] that using constant step size $\mu_k \in (0, \frac{1}{Lc_k}]$, it holds that

$$\mathbb{E}\left[r_k\left(\hat{\theta}_{k,i}\right) - r_k(\theta_k^*)\right] - \frac{\mu_k L\sigma_k^2}{2m} \leq (1 - \mu_k m)\left(\mathbb{E}\left[r_k\left(\theta_{k,i-1}\right) - r_k(\theta_k^*)\right] - \frac{\mu_k L\sigma_k^2}{2m}\right).$$

Combined with (14), we obtain

$$\mathbb{E}\left[r_k\left(\theta_{k,i}\right) - r_k(\theta_k^*)\right] - \frac{\mu_k L\sigma_k^2}{2m} \leq (1 - \mu_k m)\left(\mathbb{E}\left[r_k\left(\theta_{k,i-1}\right) - r_k(\theta_k^*)\right] - \frac{\mu_k L\sigma_k^2}{2m}\right). \tag{15}$$

Given $\mu_k \in (0, \frac{1}{Lc_k}]$, with $c_k \geq 1$, $m \leq L$, it holds that $(1 - \mu_k m) \in [0, 1)$. Applying (15) repeatedly through iteration $i \in \mathbb{N}$, we obtain

$$\mathbb{E}\left[r_k\left(\theta_{k,i}\right) - r_k(\theta_k^*)\right] \leq \frac{\mu_k L \sigma_k^2}{2m} + (1 - \mu_k m)^i \left(r_k\left(\theta_{k,0}\right) - r_k(\theta_k^*) - \frac{\mu_k L \sigma_k^2}{2m}\right)$$

$$\xrightarrow{i \to \infty} \frac{\mu_k L \sigma_k^2}{2m}.$$

This means $\theta_{k,i}$ converges towards $\theta_k^*$ with the expected regret bounded by $\frac{\mu_k L \sigma_k^2}{2m}$. $\qquad\square$

## B  Simulation Details and Supplementary Results

### B.1  Simulation details of Target Localization

The four target locations in $\mathbb{R}^2$ are: $(10.84, 10.76), (20.42, 20.26), (20.51, 10.40), (10.78, 20.30)$. Agents' locations are indicated in Figure 1a. An edge between two agents means they are neighbors. At each iteration, every agent $k$ has a noisy observation (streaming data) of the distance $\boldsymbol{d}_k(i)$ and the unit direction vector $\boldsymbol{u}_{k,i}$ pointing from $x_k$ to its target based on built-in sensors. Let $\theta_k \in \mathbb{R}^2$ denote the estimation of the target location for agent $k$, then the loss is computed as $\ell_k(\theta_{k,i}; \xi_k^i) = \|\boldsymbol{d}_k(i) - (\theta_k - x_k)^\top \boldsymbol{u}_{k,i}\|^2$, and the agent estimates $\theta_k$ using the SGD algorithm as well as the ATC diffusion algorithm with different weight assignment rules. The distance measurement data has noise variance $\sigma_{d,k}^2 \in [0.1, 0.2]$, and the unit direction vector has additive white Guassian noise with diagnoal covariance matrices $R_{u,k} = \sigma_{u,k}^2 I_2$, with $\sigma_{u,k}^2 \in [0.01, 0.1]$ for different $k$. We tune the step-sizes and forgetting factors from the interval $(0, 1)$ and find the best empirical performance by setting them to be $\mu_k = 0.1$ and $\nu_k = 0.1$ for every normal agent $k$. $\varphi_{lk}^{-1}$ and $\phi_{lk}^{-1}$ are initialized to be zero for all $l \in \mathcal{N}_k$. Byzantine agents are designed to continuously send random values for each dimension from the interval $[15, 16]$ at each iteration.

### B.2  Simulation details and supplementary results of Human Action Recognition

We randomly split the data into 75% training and 25% testing for each agent. During training, ten of the thirty agents are randomly selected to have access to much less data (about $\frac{1}{10}th$) than the other agents at each epoch. This is to model the realistic scenario in which some of the agents may have less data samples and they may learn slowly than others. We use mini-batch gradient descent with batch size of 10. We tune the step-sizes and forgetting factors from the interval $(0, 1)$ and find the best empirical performance by setting them to be $\mu_k = 0.01$ and $\nu_k = 0.05$ for every normal agent $k$. $\varphi_{lk}^{-1}$ and $\phi_{lk}^{-1}$ are initialized to be zero for all $l \in \mathcal{N}_k$. Byzantine agents are designed to send a model with very small noisy elements for each dimension from the interval $[0, 0.1]$ at each iteration.

Figure 6b shows the average *testing* loss and classification accuracy of the normal agent when 29 out of 30 agents are Byzantine (the only normal agent has access to the entire training data). Figure 5 and Figure 6a show the mean and range of the average *training* loss and classification accuracy of the normal agents in the case of no attack, with 10 random selected Byzantine agents, and with 29 Byzantine agents. In all the examples, for both training and testing, we observe that the loss-based weight assignment rule (10) outperforms the other rules as well as the non-cooperative case, with respect to the mean and range of the average loss and accuracy, which validates the result indicated by (5). Even in the extreme case in which there is only one normal agent in the network and all of its neighbors are Byzantine, the loss-based weight assignment rule (10) has the same performance as the non-cooperative case, showing its resilience to an arbitrary number of Byzantine agents.

(a) No attack                    (b) 10 Byzantine agents

Figure 5: Human Action Recognition: average training loss and accuracy for normal agents.

(a) Training                (b) Testing

Figure 6: Human Action Recognition: average training/testing loss and accuracy for normal agents with 29 Byzantine agents.

## B.3 Simulation details and supplementary results of Digit Classification

The preprocessed examples of the two datasets are given in Figure 7. The details of the CNN architecture is given in Table 1. For each group, we consider that agents have access to uneven sizes of training data. Specifically, for each agent, we randomly feed $200 - 2000$ training data and $400$ testing data from the corresponding dataset for each epoch. We use mini-batch gradient descent with batch size of $64$. We tune the step-sizes and forgetting factors from the interval $(0, 1)$ and find the best empirical performance by setting them to be $\mu_k = 0.001$ and $\nu_k = 0.05$ for every normal agent. $\varphi_{lk}^{-1}$ and $\phi_{lk}^{-1}$ are initialized to be zero for all $l \in \mathcal{N}_k$. Byzantine agents are designed to send a model with very small noisy elements for each dimension from the interval $[0, 0.1]$ at each iteration.

Since the performance of agents in the two groups diverges, we plot the results separately for the two groups. Figure 8a and Figure 8b show the average *testing* loss and classification accuracy of the normal agents in group 1 and group 2, when 8 out of 10 agents (four for each group) are Byzantine (the only normal agent in each group has access to 2000 training data).

Figure 9 and Figure 11a show the mean and range of the average *training* loss and classification accuracy of the normal agents in group 1, in the case of no attack, with 2 Byzantine agents, and with 8 Byzantine agents, which are selected randomly. Figure 10 and Figure 11b show the mean and range of the average *training* loss and classification accuracy of the normal agents in group 2, in the case of no attack, with 2 Byzantine agents, and with 8 Byzantine agents (again selected randomly). In all the examples, for both training and testing, we observe that the loss-based weight assignment rule (10) outperforms the other rules as well as the non-cooperative case, with respect to the mean and range of the average loss and accuracy, thereby validating the result indicated by (5). Even in the extreme case in which there is only one normal agent in each group and all of the other agents are Byzantine, the loss-based weight assignment rule (10) has the same performance as the non-cooperative case, showing its resilience to an arbitrary number of Byzantine agents.

Comparing the results between groups 1 and 2 reveals that cooperation is most beneficial when there is a substantial divergence in agents' learning performances. Given limited training data, agents in group 1 are able to build refined models. It is harder for agents receiving less training data in group 2 to achieve a high learning performance as the synthetic digit classification is a more challenging task than the MNIST digit classification. Using the weight assignment rule (10), those agents receiving less data (and therefore, struggling to learn a good model), are able to benefit from the cooperation with the neighbors having learned a refined model. At the same time, agents exhibiting high learning performance will not be negatively affected by such cooperation.

Table 1: CNN architecture of Digit Classification

| Layer (type) | Output Shape | Param # |
|---|---|---|
| Conv2d-1 | [-1, 32, 28, 28] | 320 |
| ReLU-2 | [-1, 32, 28, 28] | 0 |
| MaxPool2d-3 | [-1, 32, 14, 14] | 0 |
| Conv2d-4 | [-1, 64, 14, 14] | 18,496 |
| ReLU-5 | [-1, 64, 14, 14] | 0 |
| MaxPool2d-6 | [-1, 64, 7, 7] | 0 |
| Conv2d-7 | [-1, 64, 7, 7] | 36,928 |
| ReLU-8 | [-1, 64, 7, 7] | 0 |
| MaxPool2d-9 | [-1, 64, 3, 3] | 0 |
| Linear-10 | [-1, 128] | 73,856 |
| ReLU-11 | [-1, 128] | 0 |
| Linear-12 | [-1, 10] | 1,290 |

(a) MNIST

(b) Synthetic digits

Figure 7: Examples of the digit classification dataset

(a) Group 1

(b) Group 2

Figure 8: Digit Classification: average testing loss and accuracy for normal agents, with 8 Byzantine agents (four for each group).

(a) No attack

(b) 2 Byzantine agents

Figure 9: Digit Classification: average training loss and accuracy for normal agents in group 1.

(a) No attack

(b) 2 Byzantine agents

Figure 10: Digit Classification: average training loss and accuracy for normal agents in group 2.

(a) Group 1

(b) Group 2

Figure 11: Digit Classification: average training loss and accuracy for normal agents, with 8 Byzantine agents (four for each group).