[Reviews · NeurIPS 2020]

Review 1

Summary and Contributions: This work addresses distributed learning of agents, where even though each agent's training data are different, they can share their learnt model parameters over a neighborhood graph. Authors propose a weight training method that is resilient to any number of byzantine agents present in the network, that could adversarially send model updates that hamper other normal agents. The method ensures that even in the worst case, the regret of a normal agent is not more than the baseline of non-cooperative training. Their approach is validated in experiments, where they outperform model averaging that does not filter for adversaries. After reading author feedback: I thank the authors for the clarification. I maintain my positive score.

Strengths: The problem studied could be useful to the community. The core idea is intuitive, model updates from other agents are only considered if they evaluate to a lower regret on the agent's local data.

Weaknesses: Some aspects of the problem setup do not seem to be critical to the solution method. For example, the entire connectivity graph is presumably known, but no particular property of the graph (eg its laplacian matrix) is used. In the experiments, it seems all the graphs considered are complete graphs.

Correctness: I briefly checked some of the proofs in the appendix and they seemed correct to me.

Clarity: The paper is written well and is easy to follow. The digit recognition experiment was interesting and it might be better to include more description and possibly fig 8 in the main text.

Relation to Prior Work: There is adequate coverage of related work. The difference from previous work is clear. While the problem the authors are solving is different from "consensus", it would be good to include some references from the literature and highlight differences / make connections if possible. One example reference from that literature is Olfati-Saber, Reza, J. Alex Fax, and Richard M. Murray. "Consensus and cooperation in networked multi-agent systems." Proceedings of the IEEE 95.1 (2007): 215-233.

Reproducibility: Yes

Additional Feedback:


Review 2

Summary and Contributions: The paper proposes a method for performing multi-task learning (MTL) in the distributed setting when byzantine agents are present. The common approach to tackling the distributed MTL problem can fail if there is a single byzantine agent, and the authors propose an alternative online weight adjustment rule that circumvents this issue. The authors prove regret bounds for their approach. In three different sets of experiments, their algorithms converges faster than other approaches.

Strengths: The paper is well-written and reasonably easy to follow, albeit somewhat dense in terms of notation. The main result itself is interesting. The experiments presented are quite convincing.

Weaknesses: I think the authors could do a better job comparing their work to methods proposed in [5] and [16] (as described at the top of page 2). While it is clear that their approach can be resilient in a greater range of settings, I would have liked to see how their method compares to those in the settings where the methods from [5, 16] were designed to handle. A discussion of the respective theoretical results or a empirical comparison would be useful.

Correctness: The content in the main paper looks fine. I did not go through the proofs in the appendix.

Clarity: Yes. It is quite dense, but this is probably unavoidable. The trade-off is that the authors were able to provide a significant amount of intuition within the paper.

Relation to Prior Work: Mostly, but see comment under 'weakness'

Reproducibility: Yes

Additional Feedback: See comments under 'weakness'


Review 3

Summary and Contributions: This paper introduces a method for Byzantine resilient distributed multi-task learning. In particular, an online weight assignment rule is presented by exploiting the variation of the accumulated loss. The property of the approach is analyzed under the setup of convex models. The effectiveness of the approach is demonstrated with regression and classification tasks.

Strengths: A distributed MTL method is presented and some theoretical analysis is provided for the optimization.

Weaknesses: First, the scope of this paper is limited. Existing deep multi-task learning literatures are totally ignored. “An Overview of Multi-Task Learning in Deep Neural Networks, by Sebastian Ruder”. On the other hand, deep distributed learning is also not considered in this work, such as “Large Scale Distributed Deep Networks by Jeffrey Dean et al.”, “FedNAS: Federated Deep Learning via Neural Architecture Search by Chaoyang He et al.”. Both aspects should be incorporated into the paper to provide a whole picture of the development of distributed multi-task learning. Second, the proposed method is based on some strong convex assumption. Thus, when using deep multi-task representation learning way, the framework is not applicable because of the non-convex property. On the one hand, the paper should discuss the alternatives for the loss-based weight optimization in such cases. A related work is “multi-task learning using uncertainty to weigh losses for scene geometry and semantics by Alex Kendall”. On the other hand, more analysis with deep structures should be added to improve the applicability for real-world multi-task big data learning, such as “cross-stitch networks for multi-task learning”. Third, more experiments related to deep MTL should be provided to show its ability in real-world applications, especially for the digit classification. Some comparison with deep distributed learning also should be added to show the superiority of the proposed method.

Correctness: Correct under some assumptions.

Clarity: Good

Relation to Prior Work: Some important related work is not provided.

Reproducibility: Yes

Additional Feedback: After reading the feedback, I still think it is necessary to discuss the applicability for non-convex cases thoroughly, other than only providing a MNIST experiment with a simple CNN setup.


Review 4

Summary and Contributions: The paper proposes a method for distributed multi-task learning that is resilient to an arbitrary number of Byzantine agents, including the extreme case where all neighbors are Byzantine. The method is simple, theoretically sound, and has excellent computational complexity. Empirically, it appears to work well in a variety of settings, though I have some concerns about the experimental setup. Update: I have read the author response, and thank the authors for the clarification of what it means to be Byzantine.

Strengths: Despite a fairly lengthy derivation and a few approximations along the way, the idea underlying the proposed method is simple and intuitive: if a node j wants to determine if a neighbor k is trustworthy, it should evaluate the loss function using k's parameters but on j's data, and filter k out if the loss turns out to be too high. I consider the simplicity of the core idea to be a major strength of the method. Furthermore, the method works in the presence of an arbitrary number of Byzantine agents, and achieves this without requiring the user to input a tuning parameter that depends on the number of Byzantine agents (which seems to bee a weakness of previous methods). Another major strength of the proposed method, is that it automatically reduces to the non-cooperative (i.e. isolated) case when all neighbors are Byzantine. Overall, the proposed method has nice properties and is quite elegant.

Weaknesses: The experiments section did not define what it means for an agent to be Byzantine (for example, what model is used to perturb messages sent to neighbors?). Will the method still perform well if the perturbation of messages is small?

Correctness: To the best of my understanding, the derivation appears correct, though I did not check it thoroughly. The empirical methodology has a flaw, in that it does not define what it means for an agent to be Byzantine.

Clarity: The paper is easy to understand, though it is mathematically quite dense. To improve readability, it might be helpful to push some of the derviation details to the appendix.

Relation to Prior Work: I am not familiar with the literature on distributed MTL, but the related work section does lay out a decent number of baselines. My understanding so far is that the Byzantine resilience feature of the proposed method is a highly original contribution.

Reproducibility: Yes

Additional Feedback:

[Author Response · NeurIPS 2020]

We thank all the reviewers for their valuable comments and appreciation of the ideas and results presented in the paper.
We summarize the main questions from the reviewers and address them separately below.
**To Reviewer #1 Q1: Network connectivity is presumably known . . . it seems all the graphs considered are com-**
**plete graphs.** We note that the network connectivity is not assumed to be known. Agents only interact with their
local neighbors and do not know the entire network structure. There are no constraints on the network connectivity to
guarantee the convergence of the proposed algorithm. Moreover, in experiments, we have considered networks that are
**not complete** ( **Figure 1a** in the paper), which have similar experimental results to that of the complete networks.
**Q2**: **Include more description about digit classification/some refs . . .** We thank the reviewer for the useful comment
and will include further description and refs in the revision.
**To Reviewer #2 Q1: Comparison to [5, 16].** We add exper-
imental comparison of our work with [5] and [16] here ([5]

[5] fails in Byz. systems.  [16] fails when F is small.  Ours succeeds to settings in [5].

Fig.1

Fig.2
No-cooperation
algo. of [16], F = 0
algo. of [16], F = 1
algo. of [16], F = 2

Fig.3
No cooperation
Loss-based (Our method)

and this paper: 30 agents, human action recognition; [16]: 100
agents, target localization). We note that [5] considers fault-
tolerance to dropped nodes (that may stop sending message),
whereas [16] and this paper consider a more general resilience to Byz. attacks (that can send arbitrary messages). The
results show that our method is also resilient to attacks consisting of dropped nodes (Fig. 3). In contrast, [5] fails in
the Byz. systems (Fig. 1)—as the number of Byz. agents increases, test loss also increases. Since [16] has the same
Byz. setting as this paper, we omit experiments using our method in the setting of [16]. In contrast to our method,
[16] requires a user defined parameter $F$, which is the maximum number of Byz. agents in the neighborhood of a
normal agent. If the selected $F$ is smaller than the actual number of Byz. neighbors, then [16] fails (see Fig. 2, actual
maximum number of Byz. neighbors is 2, by setting $F = 0$ or $1$, [16] results in a worse learning performance/larger
MSD compared to no-cooperation). In comparison, **this paper is resilient to an arbitrary number of Byz. agents**
**and does not require the input $F$**. Besides, the time complexity for [16] is exponential in $F$, making it infeasible for
large networks and large number of Byz. neighbors, whereas **this paper has linear time complexity.**
**To Reviewer #3 Q1: Scope of the paper/Missing related work.** There is a large

R. Caruana. Multitask learning.
Task1  Task2  Task3  Task4

Ali H. Sayed, et. al. Multitask learning over graphs.

body of related work to MTL with different variations. The suggested refs by the
reviewer mostly deal with a different aspect in MTL, which is more related to transfer
learning and shares a different motivation/assumption compared to this paper (see
Fig.4,5). The first MTL setup usually assumes a known relationship between tasks
(e.g., learning depth/semantics from RGB images simultaneously since the two share
related representations), has data beforehand and learns in a fusion center. It usually

Fig.4  Inputs
MTL by Reviewer #3

Fig.5
MTL we consider

learns multiple objectives from a shared representation by sharing layers and splitting architecture in the deep NN,
e.g., sharing the first several layers with all the tasks and only the last layers are task-specific. In contrast, we
consider a network of agents that maintain separate models without sharing layers, the relationship between agents is
unknown, data is not collected centrally and agents learn in a distributed manner. These two MTL setups have different
applications: The first is widely used in Deep Learning, e.g., CV and NLP, whereas the second is naturally suited to
model distributed learning in **multi-agent systems** such as mobile phones, autonomous vehicles, and smart cities. We
also note that the suggested refs. *"An Overview . . . "*, *"MTL using uncertainty . . . "* and *"cross-stitch . . . "* are about
sharing layers/architecture of NN, which is not related to our MTL setting; *"Large scale . . . "* and *"FedNAS"* are about
distributed learning with deep models but not MTL. We can add an explanation to clarify the MTL scope of the paper.
**Q2: Convex model assumption.** Convex models are typically assumed in the ML literature for convergence analysis.
Although the analysis is based on convex models, we also used non-convex models, such as CNN in digit classification
(Table 1), and obtained experimental results that are similar to convex models. For non-convex models, the loss is
computed using the same approach as convex models and therefore, no alternative way is needed.
**Q3: Experiments related to deep MTL.** We have used a CNN for digit classification and compared our
method with others in this deep distributed MTL system, and show the superiority of the proposed method.
**To Reviewer #4 Q1: Byz. definition/Small perturbation at-**
**tack.** In the analysis, we show the convergence of the algorithm
in the presence of Byz. agents sending arbitrary messages. In
experiments, the particular messages sent by Byz. agents can be
found in Appendix B. Byz. agents send random values from the
interval $[15, 16]$ (in each dimension) in the target localization

Perturbation 0.01  Perturbation 1e-4  Perturbation 0

No cooperation
Average
Loss-based (Our method)
Distance-based   Fig.6

Fig.7

Fig.8

example, and from the interval $[0, 0.1]$ in the classification examples. We will move the description of Byz. messages
to the main context in the revision. We also provide additional experiments for small perturbation examples here (for
human action recognition, 30 agents, 10 Byz.). Results are similar to the ones in the manuscript when perturbations are
small. We also note that when perturbation is 0, the scenario degenerates to the non-attack case.
**Q2: Push derivation to . . . .** To improve readability, we will include explanation of the method in the beginning of the
derivation and move some of the derivations to the appendix in the revision, as suggested by the reviewer.

[Meta-Review · NeurIPS 2020]

There was some disagreement in the initial reviews. Three reviewers were quite positive, noting that the intuitive algorithmic ideas, interesting results and good empirical evaluation. One reviewer was more negative, with some concerns regarding the scope of the approach (in particular due to the convex assumption) and the lack of discussion/evaluation on deep MTL models. After reading the author rebuttal and further discussion among reviewers, I consider that the restriction of the analysis to convex scenarios is acceptable given that this work appears to be the first one on Byzantine MTL and that the authors provide a basic experiment in the nonconvex setting. Therefore, the paper is accepted, but I ask the authors to add a detailed discussion of the nonconvex setting and encourage them to include more complete deep learning experiments in the final version.